# Efficiency fluctuations and noise induced refrigerator-to-heater transition in information engines

Govind Paneru [1], Sandipan Dutta [1], Takahiro Sagawa[2], Tsvi Tlusty [1,3]✉ & Hyuk Kyu Pak [1,3]✉

Understanding noisy information engines is a fundamental problem of non-equilibrium physics, particularly in biomolecular systems agitated by thermal and active fluctuations in the cell. By the generalized second law of thermodynamics, the efficiency of these engines is bounded by the mutual information passing through their noisy feedback loop. Yet, direct measurement of the interplay between mutual information and energy has so far been elusive. To allow such examination, we explore here the entire phase-space of a noisy colloidal information engine, and study efficiency fluctuations due to the stochasticity of the mutual information and extracted work. We find that the average efficiency is maximal for non-zero noise level, at which the distribution of efficiency switches from bimodal to unimodal, and the stochastic efficiency often exceeds unity. We identify a line of anomalous, noise-driven equilibrium states that defines a refrigerator-to-heater transition, and test the generalized integral fluctuation theorem for continuous engines.

[1] Center for Soft and Living Matter, Institute for Basic Science (IBS), Ulsan 44919, Republic of Korea. [2] Department of Applied Physics, University of Tokyo, Tokyo 113-8656, Japan. [3] Department of Physics, Ulsan National Institute of Science and Technology, Ulsan 44919, Republic of Korea. ✉email: tsvitlusty@gmail.com; hyuk.k.pak@gmail.com

The demon envisioned by Maxwell sees gas molecules in a vessel and, by exploiting his knowledge about their motion, extracts mechanical work, apparently violating the second law of thermodynamics[1,2]. Resolving the paradox of this information engine revealed a deep link between the thermodynamic entropy of the system and the information transmitted about its microstate by the engine's feedback loop (i.e., the demon)[3–6]. But what if the information engine is noisy? (or in Maxwell's terms, the demon is a bit myopic and cannot measure the molecules precisely). In this case, one is faced with a fundamental problem of information theory: what is the effect of noise on the capacity of a communication channel to transmit information? A seminal result by Shannon is the noisy channel coding theorem: the capacity of the channel is the maximal mutual information between its input and output[7,8].

This "Maxwell meets Shannon" scenario of imperfect information engines is prevalent in non-equilibrium systems[6,9–11], especially in living systems where the signaling and perception are prone to noise[12–15]. For example, it was suggested that, by incorporating information feedback loops, the cell's signal transduction system can adapt to become more resilient to environmental noise[14]. Thus, owing to their fundamental significance, noisy information engines have been subject to several theoretical models[6,9,10,16] and experimental studies[17,18]. Most of these studies are limited to the measurement of the averaged thermodynamic quantities. For example, according to the generalized second law of stochastic thermodynamics[6], the average extracted work (or the average information conversion efficiency) of the engine is bounded by the average acquired information. However, due to their stochastic nature, the thermodynamic observables can wildly fluctuate along a single trajectory and often exceed the bounds set on the averages. Thus, one cannot decipher the magnitude and distribution of these fluctuations solely from the mean values. Therefore, we aim here to measure fluctuations in work, information, and efficiency of noisy information engines operating over a vast phase space of non-equilibrium steady states.

In advancing our understanding of the noisy information engines there remains a major obstacle: In general, one would expect the performance of the engine to depend on the channel's capacity as measured by its mutual information. Yet, the direct measurement of mutual information so far has been reported either in error-free colloidal engines[19–22], or in discrete electronic systems[17,18]. But more often the signal is noisy and continuous – as in the textbook colloidal models of stochastic thermodynamics or in ubiquitous molecular sensory systems[13] – therefore, evaluating mutual information requires measurement of the complete input–output probability distribution, which further depends on the noise distribution. Moreover, testing the fundamental limits set by non-equilibrium fluctuation theorems, such as the integral fluctuation theorem generalized for feedback systems[6,16,23], necessitates an experiment in which the magnitude and distribution of noise can be precisely controlled, which has not been achieved so far.

All these motivate us to examine the noisy information channels within an experimental setting which can directly measure, control, and vary the mutual information passing through the feedback loop. This allows us quantify the interplay between the performance of the engine and the capacity of the channel through the entire non-equilibrium phase space of the engine. To this end, we construct a cyclic Brownian information engine that is reset after each cycle of information transfer and work extraction. Such periodically reset engines and information channels are prevalent in stochastic thermodynamics[24,25], especially in living systems. For example, in molecular receptors that recurrently bind and unbind signaling ligands[26] and the main synthesis pathways of the central dogma[27].

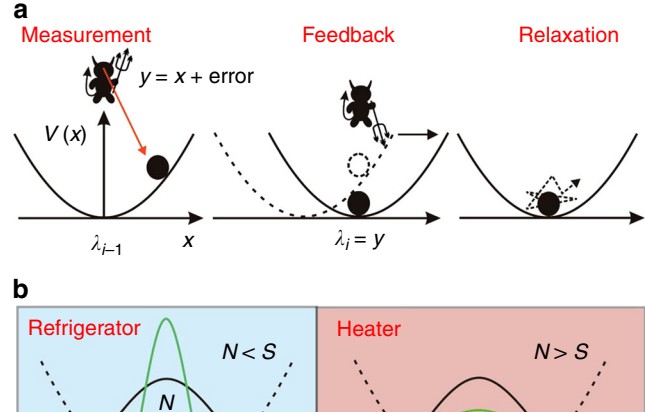

**Fig. 1 Schematics of the mutual information-fueled colloidal engine. a** Illustration of the $i$-th engine cycle. At the beginning of the $i$-th cycle, the particle is located at $x$ with respect to the potential center $\lambda_{i-1}$. The demon measures the particle position as $y = x + \text{error}$, with a Gaussian-distributed error of variance $N$. Basing on the measured $y$, the demon performs the feedback control step by instantaneously shifting the potential center to $\lambda_i = y$. The particle is then allowed to relax for a time $\tau$ in the shifted potential until the next cycle begins. **b** Illustration of the noise-induced cooling and heating regimes. The system works as a refrigerator (heater) when $N < S$ ($N > S$), where $S$ is the variance of the equilibrium position distribution (solid black) and $N$ is the variance of the noise distribution (solid green).

Our mutual information engine consists of a colloidal particle diffusing within the harmonic potential of an optical trap (Fig. 1). Each cycle begins with a practically instantaneous and error-free detection of the particle position $x$ (Fig. 1a). A Gaussian noise, of variance $N$, is added to $x$, and feedback loop (the demon) perceives this distorted value, $y = x + \text{error}$, as the particle position. The engine then responds by swiftly shifting the potential center to the perceived particle position $y$. This is followed by a relaxation step that lasts for a period $\tau$. We measured the phase space of the consequent mutual information and thermodynamic quantities such as work, heat, efficiency, and their fluctuations, as a function of its parameters $\tau$ and $N$. For finite $\tau$, we obtained a rich variety of nonequilibrium steady states stemming from the feedback-measurement interplay. Besides the usual equilibrium state obtained by large relaxation times ($\tau \to \infty$), we find also a line of noise-driven equilibrium states, at equal levels of noise and signal. Across this line, the thermodynamic quantities as well as their fluctuations change their qualitative behavior. This line also signifies the transitions of the engine from a refrigerator to a heater. We find that engines with longer cycle period $\tau$ are more efficient, as expected. But for a given $\tau$, the most efficient engine is one with a finite noise $N$ (in Maxwell's terms, a bleary-eyed demon). We show that the efficiency exhibits a transition from bimodal to unimodal distribution, and the stochastic efficiency often exceeds the bound of the generalized second law. The maximal efficiency at non-zero noise stems from the bimodal distribution of the efficiency fluctuations. Finally, we report the first examination of the validity of the generalized integral fluctuation theorem for mutual information-fueled Brownian engine.

## Results

**The mutual information engine.** The information engine consists of a colloidal particle stochastically moving within the harmonic potential $V(x, t) = (k/2)(x - \lambda(t))^2$ of an optical trap in a bath of temperature $k_B T = \beta^{-1}$ (the experimental setup is

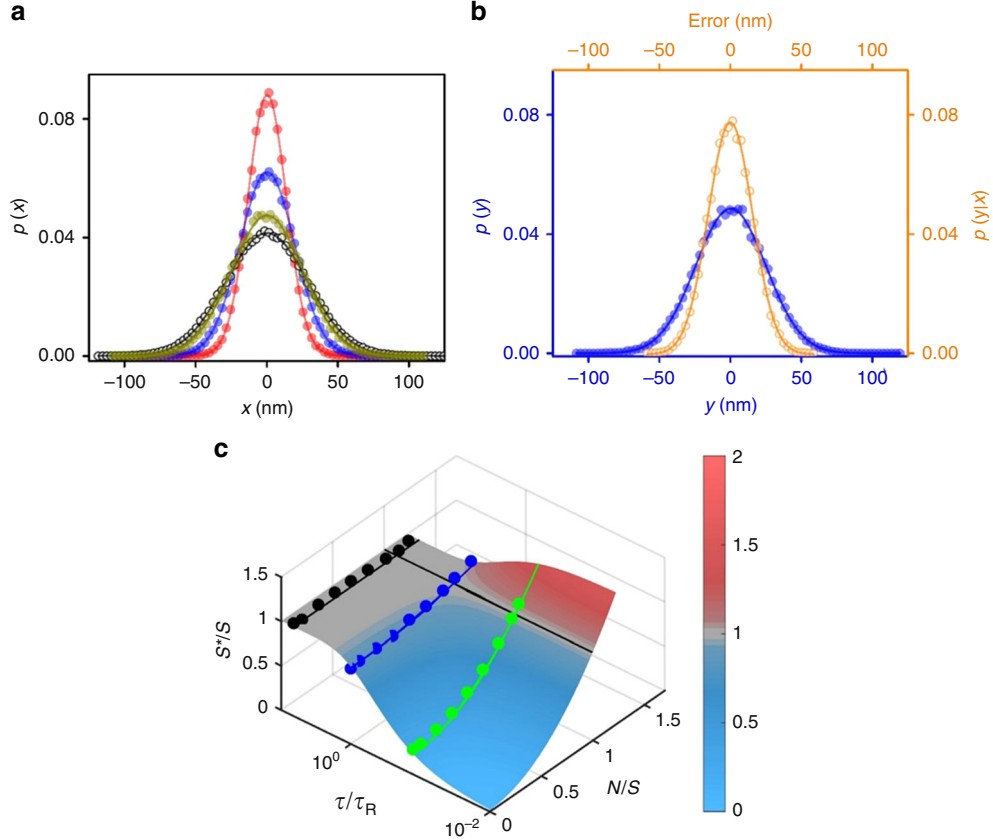

**Fig. 2 Steady state probability distribution functions. a** The measured steady state probability distribution function $p(x)$ of the true particle position $x$ for normalized noise level $N/S = 0$ (red circles), 0.28 (blue circles), and 0.69 (dark yellow circles), when cycle period $\tau$ is 0.5 ms. The distribution at thermal equilibrium without feedback is drawn in black empty circles. The solid curves are fits to Gaussians. **b** The steady state probability distribution $p(y)$ of the perceived position $y$ (blue), when a Gaussian error source $p(y \mid x)$ (orange) distorts the true position $x$, whose distribution is $p(x)$ (blue data in **a**). **c** Contour plot of the normalized steady state variance $S^*/S$ of $p(x)$ as a function of $N/S$ and $\tau/\tau_R$ using Eq. (1). The solid circles are the plot of experimentally obtained values of $S^*/S$ as a function of $N/S$ when $\tau$ is 0.5 (green circles), 3 (blue), and 20 (black) ms. The solid curves are from the model (Eq. 1). The black line parallel to $\tau/\tau_R$ axis denotes $N/S = 1$ and separates the refrigerator (blue) and heater (red) region of the engine.

expounded in Methods). Here, $k$ is the stiffness of the trap with its center at $\lambda(t)$. The colloidal particle is 2.0 μm in diameter, and its thermally agitated motion is therefore well within the over-damped low-Reynolds regime[19,28]. Without feedback, the Boltz-mann distribution of the particle position describes a Gaussian of variance $S \approx (27.4 \text{ nm})^2$ (Fig. 2a). From the variance, we calibrate the stiffness of the trap, $k = k_B T/S \approx 5.4 \text{ pN μm}^{-1}$. The timescale of the overdamped dynamics is the characteristic time it takes for the particle to relax towards equilibrium, $\tau_R = \gamma/k \approx 3.5$ ms, where $\gamma$ is the Stokes friction coefficient.

During the relaxation step, the measured particle position exhibits a time-varying Gaussian distribution, $p(x, t) = G(x, b(t), S(t))$[9,10] (See Supplementary Note 1). Here, $G$ is Gaussian distribution with time dependent center $b(t)$ and variance $S(t)$. Let us follow the dynamics of the engine along the $i$-th cycle, beginning when the particle is at position $x$ (w.r.t. the trap's center $\lambda_{i-1}$). First, the information engine detects $x$ as $y$. The noisy information channel is represented by the input-output relation $p(y|x) = G(y, x, N)$[8]. The noise broadens the distribution of the perceived position $y$ (Fig. 2b), and the particle position distribution right after the measurement becomes $p(x|y)$ (See Supplementary Note 1). Next comes the feedback step, when the engine shifts the center of the trap according to the position perceived by the noisy channel, $\lambda_{i-1} \rightarrow \lambda_i = y$ (Fig. 1a). In relative frame of reference, the trap center is fixed while the particle position $x$ is reset to $x - y$. During the last step of the protocol the

system is allowed to relax for time $\tau$ and subsequent cycle is repeated.

After many repetitions of the protocol, the engine can adequately sample the shift distribution and all probabilities reach steady state (Fig. 2). Therefore, the distribution of particle position just after the reset step is exactly the distribution of errors of the Gaussian information channel, $G(x, 0, N)$. The steady state distribution after the relaxation (at the beginning of next cycle) is then given by $p(x) = G(x, 0, S^*)$, with the variance $S^*$ (See Supplementary Note 2)

$$S^* = S + (N - S)e^{-2\tau/\tau_R}. \tag{1}$$

Figure 2a shows $p(x)$ widening with the noise level $N$, in excellent agreement with Eq. (1). Faster engines can narrow or widen the distribution, depending on the noise level $N/S$, with a minimum, $S^*/S = 1 - \exp(-2\tau/\tau_R)$, for error-free engines ($N/S = 0$) (Fig. 2c). The distribution of the measurement outcome $y$ is $p(y) = G(y, 0, S^* + N)$ (Fig. 2b).

Here, one can discern between two classes of noisy information engines (bleary-eyed demons) (Fig. 1b). The first one is the relatively accurate engine ($N < S$), that utilizes the measurement-feedback steps to narrow the distribution from a variance $S^*$ to a variance $N < S^*$ (Fig. 2c). During the relaxation step, the distribution spreads back but still remains narrower than the equilibrium one, $S^* \leq S$. At the extreme, a perfect engine shrinks the distribution down to a delta function just after the feedback,

which then expands towards equilibrium during the relaxation step. The other class is the more erroneous engines with widely distributed errors ($N > S$). By performing feedback, such engines widen the distribution to $N > S^*$. When relaxing, the distributions shrink down towards equilibrium $S^* \geq S$. The departure of $S^*$ from the equilibrium variance $S$, for any finite cycle $\tau$, can be interpreted in terms of an effective temperature of the particle, $k_B T_{\text{eff}} = k S^*$. Thus, the information engines with $N < S$ perform as refrigerators ($T_{\text{eff}}/T = S^*/S < 1$), while the ones with $N > S$ act as heaters ($T_{\text{eff}}/T > 1$), as shown in Fig. 2c. In this context, the perfect engine ($N = 0$) with $\tau = 0$ essentially operates at $T_{\text{eff}} = 0$.

**The performance of the information engine**. In the overdamped regime the kinetic energy of the particle can be ignored, so the change in the potential energy when the trap shifts, $\Delta V(x)$, is fully converted into heat and work. However, the potential is shifted much faster (within 20 μs) than the typical relaxation time such that the particle has no time to move and dissipate energy. Therefore, all the potential energy gained by the shift is converted into work. During the relaxation step, since the trap center remains fixed, no work is performed on the particle, and only heat is dissipated. Thus, the work done on the particle during each shifting of the center is $\beta W \equiv \beta \Delta V = (1/2)\beta k[(x - y)^2 - x^2]$. The average work done on the particle per cycle in steady-state $\langle \beta W \rangle$ and its fluctuation $\text{std}(\beta W)$ are (See Supplementary Note 3)

$$\langle \beta W \rangle = \frac{1}{2}\beta k \int dx\, dy\, p(x|y)p(y)\left[(x-y)^2 - x^2\right]$$
$$= -\frac{1}{2}\left(\frac{S^* - N}{S}\right) \quad (2)$$

$$\text{std}(\beta W) = \sqrt{\frac{N^2 + (S^*)^2}{2S^2}}. \quad (3)$$

The steady-state average heat supplied to the system $\langle \beta Q \rangle$ during the relaxation step is minus the average work performed on the system during the feedback, $\langle \beta Q \rangle = -\langle \beta W \rangle$ (See Supplementary Note 3). This shows that for $N < S$ the system is cooled immediately after the feedback control, and net heat flows from the reservoir to the system during the relaxation. The effective cooling decreases with increasing the error level until $N = S$ at which $\langle \beta Q \rangle = 0$. For $N > S$, the work performed on the system during the feedback is positive (heating), and net heat flows from the system to the reservoir during the relaxation. Note that our observation of cooling and heating of the system is protocol dependent. As an example, a previous theoretical work[9] shows that for a system initially in thermal equilibrium, the average extracted work $\langle -\beta W \rangle$ is always positive for an optimal protocol where the particle position is instantaneously shifted to $y \cdot S/(S + N)$.

Figure 3a shows the distribution of extracted work $-\beta W$ in several regimes of engine accuracy. A quasistatic ($\tau \gg \tau_R$) and perfect engine ($N = 0$) always extracts positive work with an average $\langle -\beta W \rangle = 0.498 \pm 0.003$, in agreement with the theoretical value 0.5 (Eq. 2). Imperfect engines ($N > 0$) sometimes make mistakes in their feedback and have non-zero probability for negative extracted work. Engines with relatively good accuracy ($N < S$) rarely make such mistakes and, on average, always extract positive work from the bath, $\langle -\beta W \rangle > 0$, performing as refrigerators. The distribution becomes symmetric for marginal engines ($N = S$) which extract no work on average, $\langle -\beta W \rangle = 0$. At the other extreme, the more erroneous engines ($N > S$), often shift the trap center too far from the particle, such that the average extracted work is negative, $\langle -\beta W \rangle < 0$, performing as

heaters. Curves of the extracted work $\langle -\beta W \rangle$ as a function of the noise level $N/S$ (Fig. 3b) agree with the theoretical prediction in Eq. (2). The maximal work $\langle -\beta W \rangle = 0.5$ is extracted by perfect engines whose cycle is long enough to reach equilibrium. While the work extracted by ultrafast engines ($\tau \rightarrow 0$) vanishes, $\langle -\beta W \rangle \rightarrow 0$, these engines extract maximum average power, $P \equiv \langle -\beta W \rangle / \tau \rightarrow (1 - N/S)/\tau_R$.

The information gain at the time of measurement is the mutual information between the particle position $x$ and the measurement outcome $y$, and is given by $I = \ln[p(x|y)/p(x)]$. Then the average steady-state mutual information $\langle I \rangle$ and its standard deviation std ($I$) are (See Supplementary Note 3)

$$\langle I \rangle = \int dx\, dy\, p(x|y)p(y)\ln\frac{p(x|y)}{p(x)}$$
$$= \frac{1}{2}\ln\left(1 + \frac{S^*}{N}\right) \quad (4)$$

$$\text{std}(I) = \sqrt{\frac{S^*}{S^* + N}}. \quad (5)$$

Since resetting the trap center erases any mutual information between $x$ and $y$, $\langle I \rangle$ is the net average information gain per cycle, $\langle \Delta I \rangle = \langle I \rangle$. The measured $\langle I \rangle$ is smaller for larger noise level $N$ and shorter cycles $\tau$, agreeing with Eq. (4) (Fig. 3c and its inset). However, it always remains greater than the average extracted work, $\langle I \rangle \geq \langle -\beta W \rangle$, in accord with the generalized second law of thermodynamics[6]. The information gained by perfect engines ($N = 0$) diverges.

Well-equilibrated engines that have enough time to relax, $\tau/\tau_R \rightarrow \infty$, have the capacity of the classical Gaussian channel[8], $\langle I \rangle = (1/2)\ln(1 + S/N)$. At the other extreme, ultrafast engines ($\tau \rightarrow 0$) still get $\langle I \rangle = (1/2)\ln(2)$ nats from each cycle (i.e. ~½ bit). This value is also the information gained by observing a particle fluctuating with variance equal to the accuracy of the measurement, $S^* = N = S$ (Fig. 3c inset). At this extreme, the feedback step does not alter the distribution leading to noise-driven-equilibrium. Finally, it follows from Eqs. (1) and (4), that increasing the cycle period will improve the information capacity of relatively accurate engines ($N < S$), but will worsen the performance of the more erroneous ones ($N > S$), consequently the amount of work extraction is suppressed, as shown in the inset of Fig. 3c.

Figure 3d shows the plot of fluctuations in extracted work std($-\beta W$) and mutual information std($I$) as a function of normalized cycle period $\tau/\tau_R$. For the more erroneous engines ($N > S$), the fluctuations in work and mutual information decrease with increasing the cycle period as expected. However, they are found to be increased for relatively accurate engines ($N < S$).

Engines operated by perfect feedback loops are not the most efficient ones (at least for the current feedback protocol). To see this, we compute and measure the average efficiency of information to work conversion, $\bar{\eta} \equiv \langle -\beta W \rangle / \langle I \rangle$ (Fig. 4a). For any given cycle period $\tau$, the most efficient engines are noisy (bleary-eyed) ones. The global maximum $\bar{\eta} \approx 0.48$ is obtained by a slower engine at $N/S \approx 0.36$, which uses only $\langle I \rangle \approx 0.67$ nats ≈ 0.97 bits of information per cycle. Retrieving more information on the particle position would have only a diminishing return. Ultrafast engines ($\tau = 0.5$ ms) are most efficient at $N/S \approx 0.26$, albeit the extracted work is very small, since the particle has little time to relax between cycles. These ultrafast engines use merely $\langle I \rangle \approx 0.5$ bits per cycle. The engines with $N > S$ exhibit negative efficiency. Our observation of maximum efficiency at finite error cannot be predicted from recently demonstrated discrete information engine[18], which shows efficiency maximum at $N/S \rightarrow 0$.

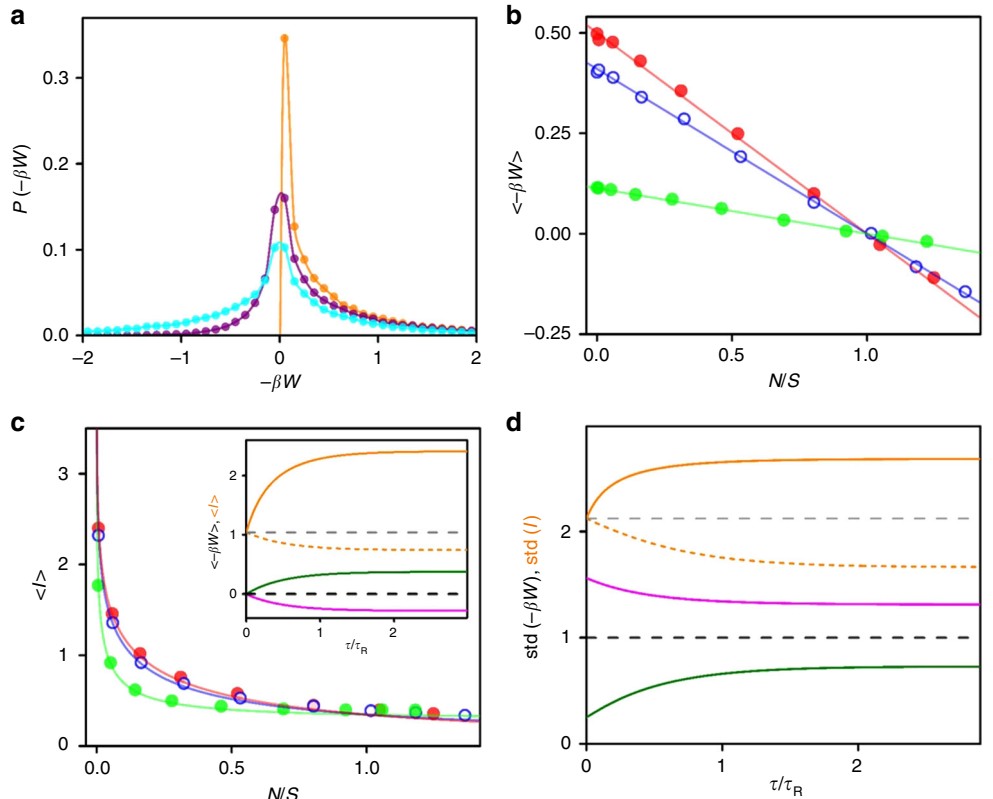

**Fig. 3 Measurement of work and mutual information. a** Experimentally measured probability distribution functions of the extracted work $-\beta W$ at steady state for cycle $\tau = 20$ ms and noise levels $N/S = 0$ (orange), 0.31 (purple) and 1.12 (cyan). The solid curves are guides to the eyes. **b** The average extracted work $\langle -\beta W \rangle$ as a function of noise level $N/S$, for $\tau = 20$ (red circles), 3 (blue), and 0.5 (green) ms. The solid lines are plot of Eq. (2). **c** The measured average mutual information in steady state $\langle I \rangle$, as a function of $N/S$ for $\tau = 20$ (red circles), 3 (blue), and 0.5 (green) ms. The solid curves are plot of Eq. (4). Inset: theoretical plot of $\langle -\beta W \rangle$ and $\langle I \rangle$ as a function of normalized cycle period $\tau/\tau_R$ (using Eqs. (2) and (4)) for $N/S = 0.5$ (olive and orange curves), 1.0 (black and grey dashed lines), and 1.25 (magenta and dashed orange curves) for $\langle -\beta W \rangle$ and $\langle I \rangle$, respectively. For better visualization, all values of mutual information are scaled up by a factor of 3. **d** Fluctuations in work std($-\beta W$) and mutual information std($I$) as a function of $\tau/\tau_R$ for like colored curves in the inset of panel **c**.

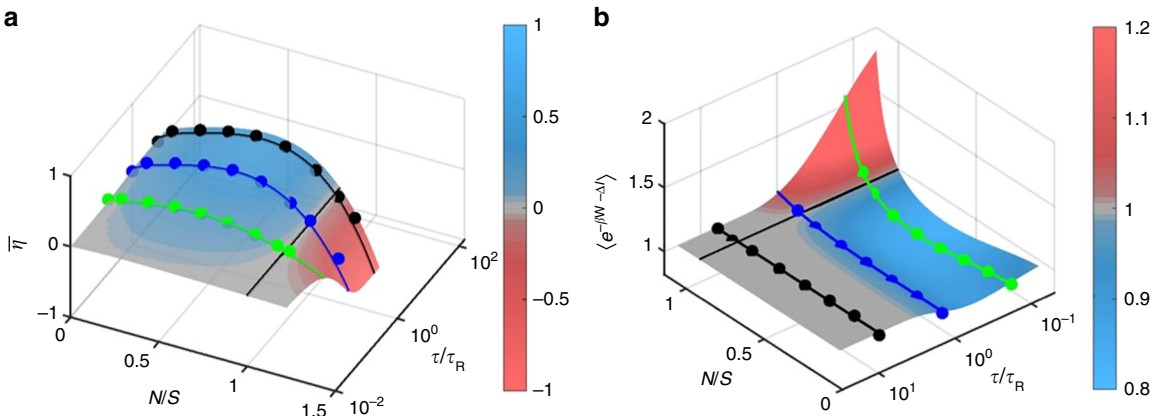

**Fig. 4 Measurement of engine efficiency and test of the generalized integral fluctuation theorem. a** The information utilization efficiency, $\bar{\eta} \equiv \langle -\beta W \rangle / \langle I \rangle$, as a function of $N/S$ and $\tau/\tau_R$. Solid circles are experimentally obtained $\bar{\eta}$ for $\tau = 20$ (black circles), 3 (blue) and 0.5 (green) ms. The solid curves are plot of the model (the ratio of Eqs. (2) and (4)). **b** Contour plot of $\langle \exp(-\beta W - \Delta I) \rangle$ as a function of $N/S$ and $\tau/\tau_R$ using Eq. (6). The solid circles are the experimental plot of $\langle \exp(-\beta W - \Delta I) \rangle$ as a function of $N/S$ for $\tau = 20$ (black circles), 3 (blue circles), and 0.5 (green circles) ms. The solid curves are obtained by plotting Eq. (6).

Interestingly, DNA recognition by transcription factors is also optimal around the regime of ~1 bit per base pair[29]. In this case, the efficiency measures how much information about the sequence can be extracted from one unit of binding energy. Thus, DNA recognition transforms energy to information most efficiently at a regime similar to that of a colloidal engine that transforms information to energy most efficiently. In both cases, the exchange between energy and information exhibits diminishing return beyond ~1 bit of information, suggesting this region as a generic optimality regime in stochastic energy-information systems.

**Test of integral fluctuation theorem**. We also test, experimentally and theoretically, the generalized integral fluctuation theorem, $\langle e^{-\beta(W-\Delta F)-\Delta I}\rangle = 1$, which is valid for system under measurement and feedback control whose initial and final states are in equilibrium[6,16], and check how far the average deviates from unity for our cyclic information engine with non-equilibrium initial and final states. The value of the average $\langle e^{-\beta(W-\Delta F)-\Delta I}\rangle$ for the current feedback protocol where $\Delta F = 0$ is equal to (See Supplementary Note 3)

$$\langle \exp(-\beta W - \Delta I)\rangle = \left[1 + \left(1 - \frac{S^*}{S}\right)\left(\frac{N+S^*}{S}\right)\right]^{-1/2}, \quad (6)$$

which becomes unity only when $S^* = S$. This condition is achieved when the engine reaches equilibrium either by relaxing for long periods, $\tau/\tau_R \to \infty$ (period-driven equilibrium), or when it mimics the equilibrium Boltzmann distribution by tuning the noise to signal, $N = S$ (noise-driven equilibrium). Experimentally (Fig. 4b), we find that $\langle e^{-\beta W-\Delta I}\rangle = 1$ regardless of error size for $\tau = 20$ ms (black circles), for which the system is fully relaxed at the end of each cycle. For finite $\tau$, $\langle e^{-\beta W-\Delta I}\rangle$ deviates from unity, even near $\tau_R$ for which the system reaches near equilibrium (blue circles). Furthermore, $\langle e^{-\beta W-\Delta I}\rangle$ is found to be always less (greater) than one in cooling (heating) region of the engine. The experimental test of a more general fluctuation theorem for total entropy production, that is valid for arbitrary initial and final states[30], $\langle e^{-\Delta S_{\text{tot}}-\Delta I}\rangle = 1$, would require the direct measurement of system entropy change, heat dissipation and mutual information along individual trajectories, which is beyond the scope of the present work.

**Efficiency fluctuations**. Our measurement shows that the average efficiency $\bar{\eta} \equiv \langle -\beta W\rangle/\langle I\rangle$ is maximal for finite error level and long cycle period. However, this maximal efficiency is practically useless due to vanishing average power $P \equiv \langle -\beta W\rangle/\tau \to 0$ in this limit. On the other hand, thermal fluctuations and fluctuations in the signal received by the detector strongly affect the operation of these microscopic engines. For example, we can show from Eq. (2) that for $N/S = 0$ and large $\tau$, the average work is maximal, $\langle -\beta W\rangle \approx 0.5$; however, it exhibits large variance, std $(-\beta W) \approx 0.7$, implying that the work obtained in individual realizations fluctuates violently around the mean. As a result, the average values alone are not sufficient for understanding and designing information engines, and one must take into account fluctuations in the thermodynamic quantities such as work, heat, and information. Typical to fluctuating systems, the most probable efficiency, at the peak of the distribution, is more informative than the average. For small systems like ours, we find that the average and the most probable values have quite distinct physical behavior.

Recent studies demonstrated that, due to the fluctuations in work and heat, the efficiency of a stochastic heat engines driven by nonequilibrium protocol is not bounded and often exceeds the limit of Carnot efficiency[31–35]. Here, we study the stochastic efficiency $\eta = -\beta W/I$ of an information engine owing to the fluctuations in work and mutual information (in the $N \le S$ regime). Figure 5a exhibits double peaks for the distribution of efficiency $p(\eta)$ for smaller noise level (orange curve) at $\tau = 3$ ms, which coalesce into a single peak (olive in Fig. 5b) at the noise level for which $\bar{\eta}$ is maximal ($N/S \approx 0.32$). For higher noise levels, $p(\eta)$ broadens (wine in Fig. 5b inset) and its peak shifts towards $\eta = 1$. Similar behavior is observed for $\tau = 0.5$ and 20 ms, except

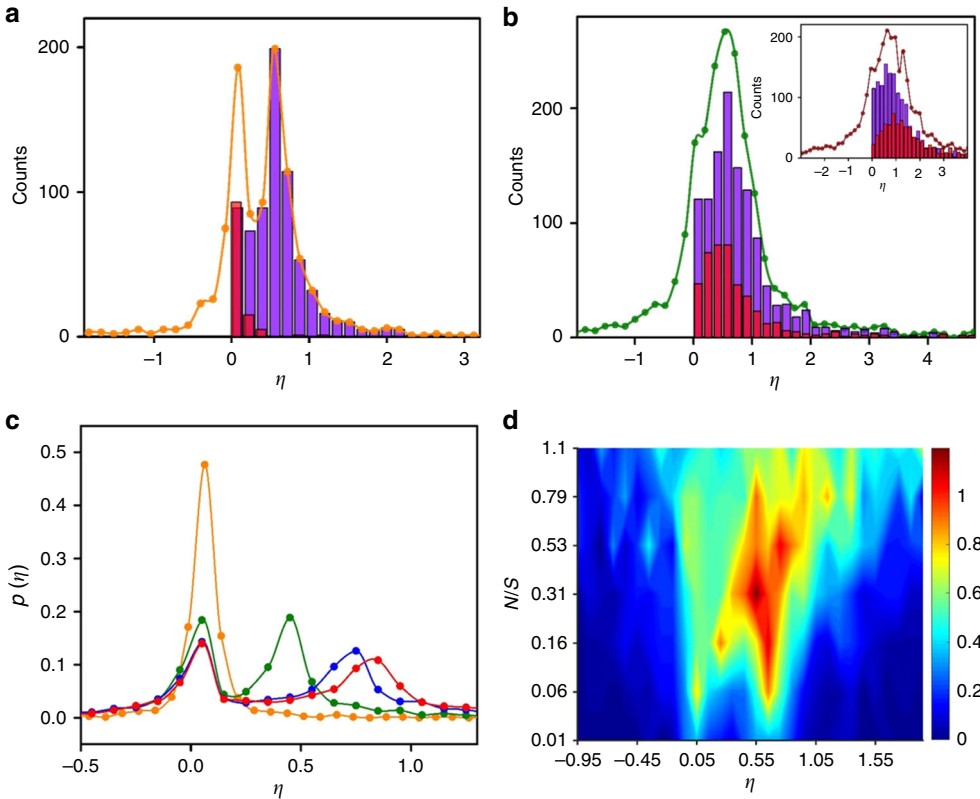

**Fig. 5 Measurement of efficiency fluctuations. a, b** Histograms of the experimentally measured efficiency $\eta = -\beta W/I$ for $\tau = 3$ ms and $N/S = 0.06$ (orange), 0.32 (olive) and 0.79 (wine in inset of **b**). The solid curves are guides to the eyes. The violet (red) bars correspond to $\beta W < 0 (\beta W > 0)$ and $I > 0$ ($I < 0$), respectively. **c** Efficiency distribution (from simulation) for $N/S = 0.034$ at four different cycle times $\tau = 0.1$ (orange), 1 (olive), 3 (blue), and 10 (red) ms. **d** Contour plot showing the fluctuations in efficiency distribution as a function of noise level $N/S$ for $\tau = 3$ ms.

that the bimodal to unimodal transition occurs at smaller $N/S$ values in engines with shorter periods.

The double peaks of the efficiency distribution $p(\eta)$ stem from the interplay between the distributions of extracted work $p(-\beta W)$ and mutual information $p(I)$, which also bifurcate with decreasing error level (Supplementary Fig. 1). At low noise levels, the peaks of $p(-\beta W)$ and $p(I)$ are well separated, giving rise to double peaks. At higher error level, they get closer, owing to sharp decrease in $I$, and eventually coalesce. In particular, the observed peaks result mainly from the negative values of $\beta W$ for which $I$ is positive. There is also contribution from positive $\beta W$ for which $I$ is negative (red bars). For larger error levels, the distribution of $\beta W$ spreads and broadens in the positive direction, and its peak shifts toward $\eta = 1$, resulting in a maximal average efficiency at a finite error level. This contribution of $\beta W > 0$ (the heater regime) to positive values of efficiency could not be predicted from the average values alone.

Figure 5c shows that the efficiency distribution exhibits a single peak near the origin for $\tau \approx 0.1$ ms, which bifurcates into bimodal distribution for a finite $\tau$, while the second peak shifts towards $\eta = 1$ for $\tau \geq \tau_R$. The origin of double peaks in our system appears consistent with recent theoretical work on stochastic heat engines[36]. However, it is noteworthy that we do not observe a minimum near $\eta = 1$; the observed peaks in our system are mainly owing to $\beta W < 0$ and $I > 0$. The derivation of an exact form of $p(\eta)$ in cyclic information engines, and in particular testing whether it asymptotically approaches universal scaling $p(\eta \to \pm\infty) \sim \eta^{-2}$ [36,37], should be an interesting future work. In a two-temperature heat engine, the efficiency distribution may exhibit peaks in the negative regime[38], whereas in our single-temperature information engine the peaks are always in the positive regime (Fig. 5c), suggesting that the information engine is capable of extracting positive work for most cycles. Interestingly, the ensemble-averaged efficiency, $\langle\eta\rangle \equiv \langle-\beta W/I\rangle$ has a global maximum near $N/S \approx 0.32$ (Fig. 5d), for which the average efficiency, $\bar{\eta} \equiv \langle-\beta W\rangle/\langle I\rangle$ is also maximal, though their values differ.

## Discussion

In conclusion, we examined the Maxwell-meets-Shannon problem by studying the mutual information-fueled Brownian engine. Information engines are special class of feedback control systems that are capable of extracting work from thermal fluctuations. Here, we incorporated the effect of feedback into the formalism of stochastic thermodynamics to realize stochastic engines operating in isothermal conditions. Our analysis shows that the engine extracts work from information about the microstate of the system without affecting the energy balance, but only the entropy balance. Thus, such feedback system affects the balance in the second law of thermodynamics but not in the first law.

By directly controlling and measuring the mutual information passing through the noisy detector used by the engine, we fully characterized the information-energy interplay of noisy Gaussian engine over a wide variety of non-equilibrium steady states both in experiment and theory. The present laws of stochastic thermodynamics cannot predict the fluctuations in the thermodynamic observables, extracted work, information, and efficiency. Nevertheless, there are recent attempts to link the fluctuations with the average dissipation by means of thermodynamic uncertainty relations[39–44].

Unlike previously reported two-bath stochastic engines[45,46], the distinctive feature of the present information engine is that one can use the noise to either heat or cool the system immersed in a single temperature bath. We obtain a refrigerator if the noise level is smaller than the signal level, or a heater otherwise. In the refrigerator ($N < S$), the system is cooled immediately after the feedback, thereby inducing a temperature difference, and net heat flows from the reservoir back to the system during relaxation with an average efficiency $\bar{\eta} = \langle\beta Q\rangle/\langle I\rangle$. In contrast, without feedback, the net heat flow is zero, and in heaters ($N > S$) heat flows from system to the reservoir. The heater and refrigerator regions in the dynamic phase diagram are separated by an anomalous, noise-driven equilibrium state along the $N = S$ line, where all thermodynamic variables and their fluctuations switch their behavior.

We find that the most efficient engines utilize merely about 0.5–1 bits of positional information per cycle. A universal feature of our information engine, irrespective of cycle period, is the transition in the distribution of efficiency fluctuation from bimodal to unimodal. Moreover, information engines with slower cycle and finite error are occasionally capable of extracting work beyond the bound set by generalized second law, for engines starting from equilibrium states.

The output power at maximum efficiency of our information engine near quasi-static regime, $\tau \sim \tau_R$, is comparable to the power of molecular motors, but about an order-of-magnitude larger than the maximal power generated of a recently reported two-temperature Brownian engine[46]. Almost all biological motors operate in noisy environment and exchange energy and information with a single-temperature bath, and hence cannot be understood on the basis two-temperature heat engine. Our study of single-temperature information engines can shed light on the underlying operation principles of these biological motors.

The generalized integral fluctuation theorem was found to be valid only when the system is fully relaxed at the beginning of each cycle or at the noise-driven equilibrium, in which the noise level is equal to that of the signal. For an arbitrary non-equilibrium steady state, it is less (greater) than unity in the refrigerator (heater) region. In the future, it would be interesting to test in our feedback system the validity of more general fluctuation theorems related to entropy production[30]. This study can be useful in designing and understanding of efficient synthetic submicron devices, as well as biological micron-scale systems, in which fluctuations of the system and the detector are inevitable.

## Methods

**Experimental**. The basic experimental setup of the colloidal information engine is shown in the Supplementary Fig. 2. A 1064-nm laser is used for trapping the particle. The laser is fed to an acoustic optical deflector (AOD) via an isolator and a beam expander. The AOD is controlled via an analog voltage controlled radio-frequency (RF) synthesizer driver. The AOD is properly mounted at the back focal plane of the objective lens so that $k$ is essentially constant while shifting the potential center. A second laser with 980 nm wavelength is used for tracking the particle position. A quadrant photo diode (QPD) is used to detect the particle position. The electrical signal from QPD is preamplified by a signal amplifier and sampled at every $\tau$ with a field-programmable gate array (FPGA) data acquisition card. The sample cell consists of highly dilute solution of 2.0 μm diameter poly-styrene particles suspended in deionized water. All experiments were carried out at $293 \pm 0.1$ K. The parameters of the trap were calibrated by fitting the probability distribution of the particle position in thermal equilibrium without a feedback process to the Boltzmann distribution, a Gaussian of variance $S \approx (27.4$ nm$)^2$. The trap stiffness is $k = k_B T/S \approx 5.4$pNμm$^{-1}$ and the characteristic relaxation time is $\tau_R = \gamma/k \approx 3.5$ ms[19]. The particle position measurement is nearly error-free with a resolution of 1 nm, and the potential center is shifted practically instantaneously within 20 μs. Each engine cycle of period $\tau$ includes three phases: measurement of the particle position, shift of the potential center, and relaxation (Fig.1a). After the position $x_i$ of the particle, relative to trap center $\lambda_i$, is measured precisely, it is distorted with random Gaussian noise of variance $N$ to get the demon-measured value $y_i$. The potential center is then shifted to $y_i$, and the particle relaxes for duration $\tau$ before the next cycle begins. In the subsequent $(i+1)$th cycle, the particle position $x_{i+1}$ is measured with respect to the shifted potential center $\lambda_i$ (the origin is reset) and the same feedback protocol is repeated. Since the origin is reset, the process does not depend on all previous measurement, even when the cycle period is smaller than the relaxation time.

## Data availability

All data associated with this study are available from corresponding author upon reasonable request.

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

## Acknowledgements
This work was supported by the taxpayers of South Korea through the Institute for Basic Science, project code IBS-R020-D1. We thank Dr. Jae Sung Lee and Prof. Albert Libchaber for insightful discussion.

## Author contributions
G.P. and H.K.P. designed the research. G.P. performed the experiment. G.P. and S.D. analyzed the data. S.D., T.S., and T.T. contributed in theory. T.T. and H.K.P. supervised the research. All authors discussed the results and implications and wrote the paper.

## Competing interests
The authors declare no competing interests.
