## [Peer Review File · Nature Communications]

Reviewers' Comments:

Reviewer #1:

Remarks to the Author:

The authors consider a Brownian particle subjected to a feedback control that allows to extract work from a single thermal reservoir. In other words they study an experimental model of a nonautonomous Maxwell demon. Since the demon measurements contain some error, the mutual information between the measurement outputs and the system quantifies the amount of work one may extract. An efficiency is introduced to quantify how close the actual conversion is from the ideal one (efficiency one). Efficiency fluctuations are also studied. Surprisingly, in finite time a finite error rate is found to improve work extraction.

Overall the paper is well written and interesting. However, I have various concerns that I would like to see addressed.

- My main concern is that the theoretical results that are used should be explained better in order to better appreciate the experimental results. I understand that most of it is in other papers, but this paper should be readable by non-experts and thus more self-contained.

- The mutual information is never clearly defined.

- What is work and which steps extract work it not discussed.

- In (4) it is said that the validity of the FT is tested. But it seems that the FT is only valid when $S^* = S$. I did not find a discussion about why this is the case. What is the meaning of the rhs of (4)?

- The discussion on efficiency fluctuations is really brief. Is the transition from simple to double peak caused by essentially the same mechanism as the one discussed in Phys. Rev. Lett. 114, 050601 (2015)? If this is indeed the case, this point should be discussed.

- I would have expected some mention of the experimental work of Bechhoefer on measuring the Shannon entropy (PNAS October 17, 2017 114 (42) 11097-11102) in the introduction. His work on experimental measurement of efficiency fluctuations (Phys. Rev. X 6, 041010 (2016)) could also be mentioned.

- Heat production in feedback controlled information erasure was always increased by errors in Diana et al. Phys. Rev. E 87, 012111 (2013). I wonder why work extraction is different and to what extent this is a system specific feature.

Reviewer #2:

Remarks to the Author:

The paper describes experimental studies and a theoretical analysis of an optically trapped colloidal particle that performs Brownian motion in water. The particle is feedback-controlled by periodic pulses that shift the center of the trap. The new position of the trap is based on the measured position of the particle. The measurement of the feedback loop brings in a Gaussian-distributed error. The goal of the paper is to measure the "work" produced by the feedback loop and to relate it to the information extracted by the feedback loop.

The work is defined as the difference between the energy of the particle in the trap potential before and after the pulse. Among the results is the demonstration of the possibility of extracting work defined this way "beyond the bound set by the generalized second law" and that the "generalized fluctuation theorem" is valid only in special circumstances.

The theory is linear and is based on the familiar solution of the Fokker-Planck equation. The obtained simple explicit expressions are discussed at length. However, a problem with the theory and the experiment is the concept of averaging. It seems that the averaging of the particle energy in the harmonic trap is not done over time, rather the energy change is calculated only during the pulse, Eq. (B20). However, the energy is fluctuating and, as seen from Eq. (B5), the variance of the fluctuations varies in time. I do not immediately see how the employed definition of the average energy and thus the average work can be justified.

The meaning and the importance of the "stochastic efficiency" needs to be explained, particularly that it seems to exist only in a limited parameter range. It is also necessary, if at all possible, to justify the relation to the DNA recognition, and more generally, to biological motors that indeed operate in the noisy environment, but nevertheless produce well-defined work.

Noisy feedback loops are common in technology and in nature. The effect of the noise is very well understood, in particular in various generators, atomic clocks, etc. Calling a feedback loop a "demon" adds hype, but not a deeper insight.

Except for the excessive hype, the scientific results are well described. On the positive side, the advantageous feature of the system is its simplicity, but then the motion of optically trapped Brownian particles has been studied in much detail. I find it important that the paper demonstrates a limited value of the extensions of thermodynamics to systems far away from thermal equilibrium. However, given the above criticism, I am not sure this is sufficient for justifying publishing the paper in Nature Communications, unless the authors can rebut the criticism. The paper may be more appropriate for a more specialized journal

Reviewer #3:

Remarks to the Author:

In their manuscript, Paneru and colleagues reported on their investigation of noisy information engines with a colloidal particle. Their approach directly built upon a recent paper from the same team [17], where they built a colloidal engine with a noise-free feedback protocol. In the current work, they added random noise to the measured position after performing a nearly error-free position measurement of a colloidal particle. So the "demon-measured" position of the particle is $y=x + \text{error}$. The authors found that the system was in a "cooling" regime if the noise was less than the variance of the equilibrium distribution, and in the "heating" regime if the noise was more than the variance of the equilibrium distribution. The authors then performed detailed study of efficiency fluctuations due to the stochasticity of the extracted work and mutual information. The experiment and data analysis seem well-conducted. However, I personally do not find this work exciting. The experimental system is basically the same as the one in their recent paper [17]. The results are also not surprising.

Response to Reviewers

Reviewer #1

The authors consider a Brownian particle subjected to a feedback control that allows to extract work from a single thermal reservoir. In other words they study an experimental model of a nonautonomous Maxwell demon. Since the demon measurements contain some error, the mutual information between the measurement outputs and the system quantifies the amount of work one may extract. An efficiency is introduced to quantify how close the actual conversion is from the ideal one (efficiency one). Efficiency fluctuations are also studied. Surprisingly, in finite time a finite error rate is found to improve work extraction.

Overall the paper is well written and interesting. However, I have various concerns that I would like to see addressed.

We thank the Referee for the encouragement, and for constructive and thoughtful comments on our manuscript. The Referee comments that our work “*is well written and interesting*” but also raised some important points, which helped us to significantly improve the manuscript. We address the points one-by-one in the following.

- My main concern is that the theoretical results that are used should be explained better in order to better appreciate the experimental results. I understand that most of it is in other papers, but this paper should be readable by non-experts and thus more self-contained.

Following the advice by the Referee, we have revised the manuscript accordingly as detailed below. Theoretical details are in the supplementary. The main text is made more self-contained by defining all the quantities like work, information and efficiency in support of the experimental work.

- The mutual information is never clearly defined.

The mutual information is now explicitly defined in the manuscript as “the mutual information between the particle position x and the measured outcome y ”. Its explicit calculation is also included in the text.

- What is work and which steps extract work it not discussed.

Following the comment by the Referee, this is now described clearly in the revised manuscript which reads: “In the overdamped regime the kinetic energy of the particle can be ignored, so the change in the potential energy when the trap shifts, $\Delta V(x)$, is fully converted into heat and work. However, the potential is shifted much faster (within $20 \mu\text{s}$) than the typical relaxation time such that the particle has no time to move and dissipate energy. Therefore, all the potential energy gained by the shift is converted into work. During the relaxation step, since the trap center remains fixed, no work is performed on the particle, and only heat is dissipated. Thus, the work done on the particle during each shifting of the potential center is $\beta W \equiv \beta \Delta V = (1/2)\beta k[(x - y)^2 - x^2]$.”

- In (4) it is said that the validity of the FT is tested. But it seems that the FT is only valid when $S^*=S$. I did not find a discussion about why this is the case. What is the meaning of the rhs of (4)?

Following this comment, we have now added discussion about the test of IFT and explained the meaning of rhs of Eq. (4) in the manuscript. Briefly, we tested experimentally and theoretically, the generalized integral fluctuation theorem, $\langle e^{-\beta(W-\Delta F)-\Delta I} \rangle = 1$, which is valid for system under measurement and feedback control whose initial and final states are in *equilibrium*, and checked how far the average deviates from unity for our cyclic information engine with *non-equilibrium* initial and final states. The RHS of Eq. (4) shows amount of deviation of $\langle e^{-\beta(W-\Delta F)-\Delta I} \rangle$ from unity as a measure of how far away the system is from equilibrium at the end of each cycle.

- The discussion on efficiency fluctuations is really brief. Is the transition from simple to double peak caused by essentially the same mechanism as the one discussed in Phys. Rev. Lett. 114, 050601 (2015)? If this is indeed the case, this point should be discussed.

We thank the Reviewer for this very helpful comment. We have added a new Section and a new plot (Fig. 5) to describe the significance of efficiency fluctuations studies and have analyzed the efficiency fluctuation result in more detail. We found that unlike the bimodal distribution of efficiency in heat engines where ($W > 0, Q < 0$) contributes predominantly to one peak and ($W < 0, Q > 0$) to another, in information engines both peaks result from ($W < 0, I > 0$) especially at low error. We have included a detailed discussion in the text. The tail part of $p(\eta)$ does not follow η^{-2} .

- I would have expected some mention of the experimental work of Bechhoefer on measuring the Shannon entropy (PNAS October 17, 2017 114 (42) 11097-11102) in the introduction. His work on experimental measurement of efficiency fluctuations (Phys. Rev. X 6, 041010 (2016)) could also be mentioned.

We thank the Referee for these very relevant references; they have been added in the current version of the manuscript.

- Heat production in feedback controlled information erasure was always increased by errors in Diana et al. Phys. Rev. E 87, 012111 (2013). I wonder why work extraction is different and to what extent this is a system specific feature.

The work extraction and heat dissipation steps are now described in detail in the manuscript. Briefly, The average heat supplied to the system $\langle \beta Q \rangle$ during the relaxation step is minus the average work performed on the system during the feedback, $\langle \beta Q \rangle = -\langle \beta W \rangle$. This shows that for $N < S$, the system is cooled immediately after the feedback control, and net heat flows from the reservoir to the system during the relaxation. The effective cooling decreases with increasing the error level until $N = S$ at which $\langle \beta Q \rangle = 0$. For $N > S$, the work performed on the system during the feedback is positive (heating), and net heat flows from the system to the reservoir during the relaxation. We provide example showing that heating and cooling are protocol dependent.

Reviewer #2

We thank the Referee for the encouragement, and for constructive, thoughtful comments on our manuscript. The Referee appreciates our work “*the scientific results are well described. On the positive side, the advantageous feature of the system is its simplicity*” but raises concern that trapped Brownian motion has been studied before. Additionally, the Reviewer comments, “*I find it important that the paper demonstrates a limited value of the extensions of thermodynamics to systems far away from thermal equilibrium.*” At the same time, the Referee raises several important concerns, which we address in detail in the following, in particular the concern regarding the novelty compared to existing literature.

- The theory is linear and is based on the familiar solution of the Fokker-Planck equation. The obtained simple explicit expressions are discussed at length. However, a problem with the theory and the experiment is the concept of averaging. It seems that the averaging of the particle energy in the harmonic trap is not done over time, rather the energy change is calculated only during the pulse, Eq. (B20). However, the energy is fluctuating and, as seen from Eq. (B5), the variance of the fluctuations varies in time. I do not immediately see how the employed definition of the average energy and thus the average work can be justified.

Following the comment by the Reviewer, we have elaborated in the revised manuscript about the work and its averaging: Each engine cycle of duration τ consists of measurement of the particle position, shifting of the trap center, and relaxation. In the overdamped regime the kinetic energy of the particle can be ignored, so the change in the potential energy when the trap shifts, $\Delta V(x)$, is fully converted into heat and work, following the first law of thermodynamics. However, in our engine, the potential is shifted much faster (within 20 μs) than the typical relaxation time such that the particle has no time to move and dissipate energy. Therefore, all the potential energy gained by the shift is converted into work. During the relaxation step, since the trap center remains fixed, no work is performed on the particle, and only heat is dissipated. Thus, the work done on the particle per cycle is equal to the work done during each shift of the potential center, $\beta W \equiv \beta \Delta V = (1/2)\beta k[(x - y)^2 - x^2]$, and the averaging is done per cycle. Since resetting makes each cycle independent, our time averaging is equivalent to averaging over cycles. Our concept of averaging is consistent with prior theoretical works (ref. 6 and 10).

- The meaning and the importance of the “stochastic efficiency” needs to be explained, particularly that it seems to exist only in a limited parameter range. It is also necessary, if at all possible, to justify the relation to the DNA recognition, and more generally, to biological motors that indeed operate in the noisy environment, but nevertheless produce well-defined work.

Following the Reviewer’s comment, the meaning and the significance of “stochastic efficiency” are explained in the revised manuscript as “Our measurement shows that the average efficiency $\bar{\eta} \equiv \langle -\beta W \rangle / \langle I \rangle$ is maximal for finite error level and long cycle period. However, this maximal efficiency is practically useless due to vanishing average power $P \equiv \langle -\beta W \rangle / \tau \rightarrow 0$ in this limit. On the other hand, thermal fluctuations and fluctuations in the detector strongly affect the operation of these microscopic engines. For example, we can calculate from Eq. (2) that for $N/S = 0$ and large τ , the

average work is maximal, $\langle -\beta W \rangle \approx 0.5$; however, it exhibits huge variance, $std(-\beta W) \approx 0.7$, implying that the $-\beta W$ value obtained in individual realizations fluctuates violently around the mean. As a result, the average values alone are not sufficient for understanding and designing information engines, and one must take into account fluctuations in the thermodynamic quantities such as work, heat, and information. Typical to fluctuating systems, the most probable efficiency, at the peak of the distribution, is more informative than the average. For small systems like ours, we find that the average and the most probable values have quite distinct physical behavior. Recent studies demonstrated that, due to the fluctuations in work and heat, the efficiency of a stochastic heat engine driven by nonequilibrium protocol is not bounded and often exceeds the limit of Carnot efficiency. Here, we study the stochastic efficiency $\eta = -\beta W/I$ of an information engine owing to the fluctuations in work and mutual information (in the $N \leq S$ regime).”

We studied efficiency fluctuations in the $N \leq S$ regime for which the engine extract positive work in average. Its study in $N > S$ may not provide important insight because the engine cannot extract positive work in average there.

Related to DNA-recognition, we added following discussion in the revised manuscript: “... the efficiency measures how much information about the sequence can be extracted from one unit of binding energy. In other words, DNA recognition transforms energy to information, but still, the most efficient regime is similar to that of the colloidal engine that transforms information to energy.”

We also cited recent theoretical work by Ito and Sagawa (ref 14) that combines Maxwell’s demon and signal transduction in living cells.

- Noisy feedback loops are common in technology and in nature. The effect of the noise is very well understood, in particular in various generators, atomic clocks, etc. Calling a feedback loop a “demon” adds hype, but not a deeper insight.

We utterly agree with the Reviewer that the notion of “demons” is equivalent to these of information engines or feedback loop. Therefore, following the Reviewer’s advice we removed about 90% of the references to demons in the text and replaced them by information engines or feedback loops. We only kept the historical context in introduction, since the concepts of Maxwell demon and information engines are intertwined, and some readers are used to this terminology.

- Except for the excessive hype, the scientific results are well described. On the positive side, the advantageous feature of the system is its simplicity, but then the motion of optically trapped Brownian particles has been studied in much detail. I find it important that the paper demonstrates a limited value of the extensions of thermodynamics to systems far away from thermal equilibrium. However, given the above criticism, I am not sure this is sufficient for justifying publishing the paper in Nature Communications, unless the authors can rebut the criticism.

We thank the Reviewer for the positive assessment of the work. The important points of criticism raised by the Reviewer are answered in detail above. Indeed, optically trapped Brownian particles were studied

before, so it is important to highlight the significance of the present work: We have proposed an exactly solvable continuous model of information engine. We show the first direct measurement of mutual information in nonequilibrium steady state and experimental verification of generalized integral fluctuation theorem. We demonstrate that measurement related nonequilibrium introduces unusual thermodynamic features, including the possibility of both heating and cooling regimes. Our studies of fluctuations in work, mutual information and efficiency are new and provide deeper and general insight into information engines. Particularly, the observation of bimodality of efficiency, and understanding its origin would be of broader interest to information theorists, nonequilibrium physicists and quantitative biologists. Additionally, most existing models are discrete, so our work would draw out the differences in physics with discrete systems.

Reviewer #3

We thank the Referee for the thoughtful comments on our manuscript. The Reviewer comments on our work that “*experiment and data analysis seem well-conducted*”. But at the same time has concerns about the similarity of the experimental setup with our prior work and, in general, does not find the work “*exciting*” or the results “*surprising*”.

We thank the Reviewer for the opportunity to elaborate on the significance of the work, which we also further highlight in the revised manuscript. We also refer the Reviewer the report of Reviewer #1 who finds that the “*the paper is well written and interesting*” and specifically comments that “*Surprisingly, in finite time a finite error rate is found to improve work extraction.*” This and other feedback we received from experts in the field make us believe that the results are indeed striking and provide deeper understanding of information engine, as suggested by the list of our chief findings:

- (1) The entire phase space (error level and cycle period) of the engine is explored (Figures 2b, 4a and 4b), to reveal a topology of T-like equilibrium region (gray), dividing the diagram into refrigerator (blue) and heater (red) phases. No previous studies showed this topology, especially the anomalous, measurement-driven equilibrium line (the leg of the “T”).
- (2) We studied fluctuations in work, information and efficiency, which are shown to exhibit a transition from bimodal to unimodal distribution. Of course, this transition could not been observed in previous studies that looked merely at mean values. We understand the origin of bimodal to unimodal transition (as suggested by the first Reviewer). We also pointed out that the tail part of $p(\eta)$ does not follow the previously predicted (Ref 36, 37) universal power law, $p(\eta \rightarrow \pm\infty) \sim \eta^{-2}$.
- (3) The measurement-driven equilibrium (which occurs for all cycle periods, not only long ones as the usual equilibrium) is also a transition line where the behavior of the main thermodynamic observable changes.
- (4) The connection between the maximal efficiency and the stochastic fluctuations of the information engine is observed and its origin is understood.
- (5) Measurement of mutual information for continuous systems – which is much more challenging than discrete systems – is reported here for the first time. This allows us to also perform first experimental verification of the integral fluctuation theorem for continuous systems and show where it is violated.
- (6) Finally, we note that our single-bath information engine extracts higher power than recently realized two-bath heat engines, and is much more relevant to biological motors, which of course operate within a single heat bath.

The Reviewer raises concerns about possible similarity to our previous work (PRL **120**, 020601 (2018)). Therefore, we would like to stress that the motivation of our prior work (Ref [19] in revised manuscript) was realization of an *error-free* information engine. Our prior work sheds light on the understanding the thermodynamics of a colloidal particle under feedback control for the case of absolute irreversibility (PRE 90, 042110 (2014)). In addition, our prior work considers slower information engine so that the system relax to thermal equilibrium at the end of each cycle. In present work, we examine a very different regime of *noisy* information engines whose entire phase space is explored. Thus, our current work on noisy information engine is very distinct from the prior work, both theoretically and experimentally. Moreover, we explore here the nonequilibrium driving of reversible engines whose physics is fundamentally

different, in the sense that the second law of thermodynamics and the IFT in noisy information engines exclude error-free measurements (absolute irreversibility) due to divergent of entropy production. We note that these basic differences are apparent from the results as detailed in the text and figures. Finally, our present study of *noisy* information engines is important in understanding thermodynamics of many non-equilibrium systems, especially in living systems where signaling and perception are prone to noise.

Reviewers' Comments:

Reviewer #1:

Remarks to the Author:

The authors have carefully and convincingly addressed my comments as well as those by the other two referees. The paper is now much more self-contained and easy to read. In my opinion the results reported in the paper are new and interesting and demonstrate the profound interplay between nonequilibrium thermodynamics and information theory. I now enthusiastically recommend publication in Nature Communications.

Reviewer #2:

Remarks to the Author:

The paper has been significantly improved. It has been made closer to physics. However, there remain some major problems that need to be addressed before a decision about recommending the paper for publication can be made.

It is still hard to relate the work to what is commonly known. How is the "information engine" related to the standard feedback control? This is probably the most common way of control in science and technology. Does the paper describe a general means of making feedback control better or optimizing it? In what respect, if so? What general features of the feedback control are established? The only distinction I can see is that the control studied in the paper is applied periodically. Is this advantageous?

I think one of the most important results of the paper is that it shows that the "laws" of "nonequilibrium thermodynamics" are not laws. This must be articulated very clearly.

Small comments:

I am confused by the statement that, "DNA recognition transforms energy to information, but still, the most efficient regime is similar to that of the colloidal engine that transforms information to energy." However, the paper says that not much work is extracted in this regime, therefore it is not particularly useful. Is the reference to DNA relevant?

I find it confusing that the efficiency can exceed unity. Doesn't it show that the quantity called "efficiency" is ill-defined?

Referee 1

The authors have carefully and convincingly addressed my comments as well as those by the other two referees. The paper is now much more self-contained and easy to read. In my opinion the results reported in the paper are new and interesting and demonstrate the profound interplay between nonequilibrium thermodynamics and information theory. I now enthusiastically recommend publication in Nature Communications.

We thank the Referee for the encouraging comments and for assessing that our results are “*new and interesting and demonstrate the profound interplay between nonequilibrium thermodynamics and information theory.*”

Referee 2

The paper has been significantly improved. It has been made closer to physics. However, there remain some major problems that need to be addressed before a decision about recommending the paper for publication can be made.

We thank the Referee for the positive assessment of our paper. The remaining concerns are addressed point-by-point below and in the enclosed revised manuscript.

(1) It is still hard to relate the work to what is commonly known. How is the “information engine” related to the standard feedback control? This is probably the most common way of control in science and technology.

Indeed, the Referee makes the correct observation that information engines are also feedback-control systems. In the revised manuscript, we explain the special properties of this class of feedback systems (p.9):

“Information engines are special class of feedback control systems that are capable of extracting work from thermal fluctuations. Here, we incorporated the effect of feedback into the formalism of stochastic thermodynamics to realize stochastic engines operating in isothermal conditions. Our analysis shows that the engine extracts work from information about the microstate of the system without affecting the energy balance, but only the entropy balance. Thus, such feedback system affect the balance in the second law of thermodynamics but not in the first law.”

Does the paper describe a general means of making feedback control better or optimizing it? In what respect, if so? What general features of the feedback control are established?

As the Referee rightfully comments, our study reveals certain optimality features of this feedback-control system. We study the performance of information engine as a function of cycle period and noise level and find the following: The extracted work is maximal for slower and error-free engine for which the mutual information diverges and most information is wasted during the relaxation phase. However, the maximal

work for slower engine is somewhat useless because the engine's power vanishes for very long cycles. On the other hand, we find that the feedback control is optimal for a finite cycle period ($\tau \sim \tau_R$) and finite error level for which the extracted work, power and efficiency are significantly larger (p. 9 in the revised manuscript)

“The output power at maximum efficiency of our information engine near quasi-static regime, $\tau \sim \tau_R$, is comparable to the power of molecular motors, but about an order-of-magnitude larger than the maximal power generated of a recently reported two-temperature Brownian engine⁴⁶.”

The only distinction I can see is that the control studied in the paper is applied periodically. Is this advantageous?

This is an interesting point by the Referee: Indeed, periodic feedback (i.e. reset) is an efficient way of utilizing the information acquired at the time of measurement, resulting in maximum work extraction. In addition, the periodic engine erases the unused information during the reset, such that the memory is not saturated and the system reaches to nonequilibrium steady state with vanishing free energy and entropy changes, $DS_{sys} = DF = 0$.

(2). I think one of the most important results of the paper is that it shows that the “laws” of “nonequilibrium thermodynamics” are not laws. This must be articulated very clearly.

Indeed, the state of affairs in non-equilibrium and equilibrium statistical mechanics are quite different. For example, the first law of thermodynamics is valid along single trajectory; however, the second law is valid on an average as predicted by the fluctuation theorems. We made this clear in the revised manuscript (p.1):

“For example, according to the generalized second law of stochastic thermodynamics⁶, the average extracted work (or the average information conversion efficiency) of the engine is bounded by the average acquired information. However, due to their stochastic nature, the thermodynamic observables can wildly fluctuate along a single trajectory and often exceed the bounds set on the averages. Thus, one cannot decipher the magnitude and distribution of these fluctuations solely from the mean values.”

And also in conclusion (p.9)

“The present laws of stochastic thermodynamics cannot predict the fluctuations in the thermodynamic observables, extracted work, information, and efficiency. Nevertheless, there are recent attempts to link the fluctuations with the average dissipation by means of thermodynamic uncertainty relations³⁹⁻⁴⁴.”

(3). Small comments:

I am confused by the statement that, “DNA recognition transforms energy to information, but still, the most efficient regime is similar to that of the colloidal engine that transforms information to energy.” However, the paper says that not much work is extracted in this regime, therefore it is not particularly useful. Is the reference to DNA relevant?

The engine extracts maximal work in the *slow* and noisy regime, where it utilizes ~ 0.97 bits of information per cycle. The point is that the DNA recognition by transcription factors is also optimal around a similar regime of ~ 1 bit per bp, corresponding to the maximum work extraction of the information engine. Small amount work is extracted in the *fastest* engines that uses ~ 0.5 bits of information per cycle. So the DNA recognition corresponds to the slow, efficient engines. This confusion is clarified in the revised manuscript (p.7):

“Interestingly, DNA recognition by transcription factors is also optimal around the regime of ~ 1 bit per base pair²⁹. In this case, the efficiency measures how much information about the sequence can be extracted from one unit of binding energy. Thus, DNA recognition transforms energy to information most efficiently at a regime similar to that of a colloidal engine that transforms information to energy most efficiently. In both cases, the exchange between energy and information exhibits diminishing return beyond ~ 1 bit of information, suggesting this region as a generic optimality regime in stochastic energy-information systems. “

(4). I find it confusing that the efficiency can exceed unity. Doesn't it show that the quantity called “efficiency” is ill-defined?

Indeed, it is important to note the essential difference between stochastic and average efficiency. The traditional definition of engine efficiency is $\bar{\eta} \equiv \langle -\beta W \rangle / \langle I \rangle$, whereas the stochastic efficiency is $\eta = -\beta W / I$. Owing to its stochastic nature, the stochastic efficiency can exceed unity due to the strong fluctuations in both information and work. An interesting point of the present study is that stochastic efficiency provides deeper insight than the average value in stochastic engines where fluctuations in thermodynamic observables such as work, heat and information are ubiquitous and large in magnitude relative to the averages.

Reviewers' Comments:

Reviewer #2:

Remarks to the Author:

The paper has been further improved. The main point, i.e., the demonstration of the limited value of the "laws" of stochastic thermodynamics, is now made clear. The relation to the standard feedback control systems is also spelled out more clearly, although generally I would prefer to see a comparison with any standard feedback control system, which operates in the exactly same conditions: fluctuations in the system and imprecision of the measurement. It seems to me that the concept of "extracting work from thermal fluctuations" strongly depends on the definition of work. Overall, I do not see a positive contribution to the theory of feedback control. However, demonstrating limitations of stochastic thermodynamics warrants publication, and I recommend publishing the paper.